# LEARNING TO EFFICIENTLY SAMPLE FROM DIFFUSION PROBABILISTIC MODELS

## ABSTRACT

Denoising Diffusion Probabilistic Models (DDPMs) have emerged as a powerful family of generative models that, yielding high-fidelity samples and competitive log-likelihoods across a range of domains, including image and speech synthesis. Key advantages of DDPMs include ease of training, in contrast to generative adversarial networks, and speed of generation, in contrast to autoregressive models. However, DDPMs typically require hundreds-to-thousands of steps to generate a high fidelity sample, making them prohibitively expensive for high dimensional problems. Fortunately, DDPMs allow trading generation speed for sample quality through adjusting the number of refinement steps during inference. Prior work has been successful in improving generation speed through handcrafting the time schedule through trial and error. We instead view the selection of the inference time schedules as an optimization problem, and show that, with a simple dynamic programming algorithm, one can find the log-likelihood-optimal discrete time schedules for any pre-trained DDPM. Our method exploits the fact that the evidence lower bound (ELBO) can be decomposed into separate KL divergence terms, and given any computation budget, we discover the time schedule that maximizes the training ELBO exactly. Our method is efficient, has no hyper-parameters of its own, and can be applied to any pre-trained DDPM with no retraining. We discover inference time schedules requiring as few as 32 refinement steps, while sacrificing less than 0.1 bits per dimension compared to the default 4,000 steps used on an ImageNet 64x64 model.

## 1 INTRODUCTION

Denoising Diffusion Probabilistic Models (DDPMs) have emerged as a powerful class of generative models (Sohl-Dickstein et al., 2015; Ho et al., 2020). DDPMs model the data distribution through an iterative denoising process, and have been applied successfully to a variety of applications, including unconditional image generation (Song & Ermon, 2019; Ho et al., 2020; Song et al., 2021; Nichol & Dhariwal, 2021), shape generation (Cai et al., 2020), text-to-speech (Chen et al., 2021; Kong et al., 2020) and single image super-resolution (Saharia et al., 2021; Li et al., 2021).

DDPMs are easy to train, featuring a simple denoising objective (Ho et al., 2020) with noise schedules that successfully transfer across different models and datasets. This contrasts to Generative Adversarial Networks (GANs) (Goodfellow et al., 2014), which require an inner-outer loop optimization procedure that often entails instability and requires careful hyperparameter tuning. DDPMs also admit a simple non-autoregressive inference process; this contrasts to autoregressive models with often prohibitive computational costs on high dimensional data. The DDPM inference process starts with samples from the corresponding prior noise distribution (e.g., standard Gaussian), and iteratively denoises the samples under the fixed noise schedule. However, DDPMs often need hundreds-to-thousands of denoising steps (each involving a feedforward pass of a large neural network) to achieve strong results. While this process is still much faster than autoregressive models, this is still often computationally prohibitive, especially when modeling high dimensional data.

There has been much recent work focused on improving the sampling speed of DDPMs. WaveGrad (Chen et al., 2021) introduced a manually crafted schedule requiring only 6 refinement steps; however, this schedule seems to be only applicable to the vocoding task where there is a very strong conditioning signal. Denoising Diffusion Implicit Models (DDIMs) (Song et al., 2020) accelerate sampling from

pre-trained DDPMs by relying on a family of non-Markovian processes. They accelerate the generative process through taking multiple steps in the diffusion process. However, DDIMs sacrifice the ability to compute log-likelihoods. Nichol & Dhariwal (2021) also explored the use of ancestral sampling with a subsequence of the original denoising steps, trying both a uniform stride and other hand-crafted strides. San-Roman et al. (2021) improve few-step sampling further by training a separate model after training a DDPM to estimate the level of noise, and modifying inference to dynamically adjust the noise schedule at every step to match the predicted noise level.

All these fast-sampling techniques rely on a key property of DDPMs – there is a decoupling between the training and inference schedule. The training schedule need not be the same as the inference schedule, e.g., a diffusion model trained to use 1000 steps may actually use only 10 steps during inference. This decoupling characteristic is typically not found in other generative models. In past work, the choice of inference schedule was often considered a hyperpameter selection problem, and often selected via intuition or extensive hyperparmeter exploration (Chen et al., 2021). In this work, we view the choice of the timesteps of the inference schedule (which we just call an *inference path*) as an independent optimization problem, wherein we attempt to learn the best schedule. Our approach relies on the observation that we can solve this optimization problem with dynamic programming. Given a fixed budget of $K$ refinement steps and a pre-trained DDPM, we find the set of timesteps that maximizes the corresponding evidence lower bound (ELBO). As an optimization objective, the ELBO has a key *decomposability* property: the total ELBO is the sum of individual KL terms, and for any two inference paths, if the timesteps $(s, t)$ contiguously occur in both, they share a common KL term, therefore admitting memoization (see Section 4.1 for a precise definition).

Our main contributions are the following:

- We introduce a method that that finds the likelihood-optimal inference paths with a simple dynamic programming algorithm for *all* possible computation budgets of $K$ refinement steps. The algorithm searches over $T > K$ timesteps, only requiring $\mathcal{O}(T)$ neural network forward passes. It only needs to be applied once to a pre-trained DDPM, does not require training or retraining a DDPM, and is applicable to both time-discrete and time-continuous DDPMs.
- We experiment with DDPM models from prior work. On both $L_{\text{simple}}$ CIFAR10 and $L_{\text{hybrid}}$ ImageNet 64x64, we discover schedules which require only 32 refinement steps, yet sacrifice only 0.1 bits per dimension compared to their original counterparts with 1,000 and 4,000 steps, respectively.
- We show that our method can be applied to any *decomposable* set of objectives. In particular, optimizing a *reweighted* ELBO can favourably bias our algorithm towards solutions with better FID scores, as we find that optimizing the exact variational lower bound may lead to worse FID scores, which is consistent with prior work on unconditional image generation.

## 2   BACKGROUND ON DENOISING DIFFUSION PROBABILISTIC MODELS

Denoising Diffusion Probabilistic Models (DDPMs) (Ho et al., 2020; Sohl-Dickstein et al., 2015) are defined in terms of a *forward* Markovian diffusion process $q$ and a learned reverse process $p_\theta$. The forward diffusion process gradually adds Gaussian noise to a data point $\boldsymbol{x}_0$ through $T$ iterations,

$$q(\boldsymbol{x}_{1:T} \mid \boldsymbol{x}_0) = \prod_{t=1}^{T} q(\boldsymbol{x}_t \mid \boldsymbol{x}_{t-1}), \tag{1}$$

$$q(\boldsymbol{x}_t \mid \boldsymbol{x}_{t-1}) = \mathcal{N}(\boldsymbol{x}_t \mid \sqrt{\alpha_t}\,\boldsymbol{x}_{t-1}, (1 - \alpha_t)\boldsymbol{I}), \tag{2}$$

where the scalar parameters $\alpha_{1:T}$ determine the variance of the noise added at each diffusion step, subject to $0 < \alpha_t < 1$. The learned reverse process aims to model $q(\boldsymbol{x}_0)$ by inverting the forward process, gradually removing noise from signal starting from pure Gaussian noise $\boldsymbol{x}_T$,

$$p(\boldsymbol{x}_T) = \mathcal{N}(\boldsymbol{x}_T \mid \boldsymbol{0}, \boldsymbol{I}) \tag{3}$$

$$p_\theta(\boldsymbol{x}_{0:T}) = p(\boldsymbol{x}_T) \prod_{t=1}^{T} p_\theta(\boldsymbol{x}_{t-1} \mid \boldsymbol{x}_t) \tag{4}$$

$$p_\theta(\boldsymbol{x}_{t-1} \mid \boldsymbol{x}_t) = \mathcal{N}(\boldsymbol{x}_{t-1} \mid \mu_\theta(\boldsymbol{x}_t, t), \sigma_t^2 \boldsymbol{I}). \tag{5}$$

The parameters of the reverse process can be optimized by maximizing the following variational lower bound on the training set:

$$\mathbb{E}_q \log p(\boldsymbol{x}_0) \geq \mathbb{E}_q \left[ \log p_\theta(\boldsymbol{x}_0 | \boldsymbol{x}_1) - \sum_{t=2}^{T} D_{\text{KL}}\big(q(\boldsymbol{x}_{t-1} | \boldsymbol{x}_t, \boldsymbol{x}_0) \| p_\theta(\boldsymbol{x}_{t-1} | \boldsymbol{x}_t)\big) - L_T(\boldsymbol{x}_0) \right] \tag{6}$$

where $L_T(\boldsymbol{x}_0) = D_{\mathrm{KL}}\big(q(\boldsymbol{x}_T|\boldsymbol{x}_0) \,\|\, p(\boldsymbol{x}_T)\big)$. Nichol & Dhariwal (2021) have demonstrated that training DDPMs by maximizing the ELBO yields competitive log-likelihood scores on both CIFAR-10 and ImageNet $64\times64$ achieving 2.94 and 3.53 bits per dimension respectively.

Two notable properties of Gaussian diffusion process that help formulate DDPMs tractably and efficiently include:

$$q(\boldsymbol{x}_t \mid \boldsymbol{x}_0) = \mathcal{N}(\boldsymbol{x}_t \mid \sqrt{\gamma_t}\,\boldsymbol{x}_0, (1-\gamma_t)\boldsymbol{I})\,, \qquad \text{where } \gamma_t = \prod_{i=1}^{t} \alpha_i\,, \tag{7}$$

$$q(\boldsymbol{x}_{t-1} \mid \boldsymbol{x}_0, \boldsymbol{x}_t) = \mathcal{N}\left(\boldsymbol{x}_{t-1} \,\bigg|\, \frac{\sqrt{\gamma_{t-1}}\,(1-\alpha_t)\boldsymbol{x}_0 + \sqrt{\alpha_t}\,(1-\gamma_{t-1})\boldsymbol{x}_t}{1-\gamma_t}, \frac{(1-\gamma_{t-1})(1-\alpha_t)}{1-\gamma_t}\boldsymbol{I}\right) \tag{8}$$

Given the marginal distribution of $\boldsymbol{x}_t$ given $\boldsymbol{x}_0$ in (7), one can sample from the $q(\boldsymbol{x}_t \mid \boldsymbol{x}_0)$ independently for different $t$ and perform SGD on a randomly chosen KL term in (6). Furthermore, given that the posterior distribution of $\boldsymbol{x}_{t-1}$ given $\boldsymbol{x}_t$ and $\boldsymbol{x}_0$ is Gaussian, one can compute each KL term in (6) between two Gaussians in closed form and avoid high variance Monte Carlo estimation.

## 3   LINKING DDPMs TO CONTINUOUS TIME AFFINE DIFFUSION PROCESSES

Before describing our approach to efficiently sampling from DDPMs, it is helpful to link DDPMs to continuous time *affine* diffusion processes, as it shows the compatibility of our approach to both time-discrete and time-continuous DDPMs (Song et al., 2021; Kingma et al., 2021). Let $\boldsymbol{x}_0 \sim q(\boldsymbol{x}_0)$ denote a data point drawn from the empirical distribution of interest and let $q(\boldsymbol{x}_t|\boldsymbol{x}_0)$ denote a stochastic process for $t \in [0,1]$ defined through an affine diffusion process through the following stochastic differential equation (SDE):

$$dX_t = f_{\mathrm{sde}}(t)X_t dt + g_{\mathrm{sde}}(t)dB_t\,, \tag{9}$$

where $f_{\mathrm{sde}}, g_{\mathrm{sde}} : [0,1] \to [0,1]$ are integrable functions satisfying $f_{\mathrm{sde}}(0) = 1$ and $g_{\mathrm{sde}}(0) = 0$.

Following Särkkä & Solin (2019) (section 6.1), we can compute the exact marginals $q(\boldsymbol{x}_t|\boldsymbol{x}_s)$ for any $0 \le s < t \le 1$. We get:

$$q(\boldsymbol{x}_t \mid \boldsymbol{x}_s) = \mathcal{N}\left(\boldsymbol{x}_t \,\bigg|\, \psi(t,s)\boldsymbol{x}_s, \left(\int_s^t \psi(t,u)^2 g(u)^2 du\right)\boldsymbol{I}\right) \tag{10}$$

where $\psi(t,s) = \exp\int_s^t f(u)du$. Since these integrals are difficult to work with, we instead propose (in parallel to Kingma et al. (2021)) to *define* the marginals directly:

$$q(\boldsymbol{x}_t \mid \boldsymbol{x}_0) = \mathcal{N}(\boldsymbol{x}_t \mid f(t)\boldsymbol{x}_0, g(t)^2\boldsymbol{I}) \tag{11}$$

where $f, g : [0,1] \to [0,1]$ are differentiable, monotonic functions satisfying $f(0) = 1, f(1) = 0, g(0) = 0, g(1) = 1$. Then, by implicit differentiation it follows that the corresponding diffusion is

$$dX_t = \frac{f'(t)}{f(t)}X_t dt + \sqrt{2g(t)\left(g'(t) - \frac{f'(t)g(t)}{f(t)}\right)}dB_t\,. \tag{12}$$

We provide a proof for Equation 12 in the appendix (A.1). To complete our formulation, let $f_{ts} = \frac{f(t)}{f(s)}$ and $g_{ts} = \sqrt{g(t)^2 - f_{ts}^2 g(s)^2}$. Then, it follows that for any $0 < s < t \le 1$ we have that

$$q(\boldsymbol{x}_t \mid \boldsymbol{x}_s) \;=\; \mathcal{N}\left(\boldsymbol{x}_t \mid f_{ts}\boldsymbol{x}_s, g_{ts}^2\boldsymbol{I}\right)\,, \tag{13}$$

$$q(\boldsymbol{x}_s \mid \boldsymbol{x}_t, \boldsymbol{x}_0) \;=\; \mathcal{N}\left(\boldsymbol{x}_s \,\bigg|\, \frac{1}{g_{t0}^2}(f_{s0}g_{ts}^2\boldsymbol{x}_0 + f_{ts}g_{s0}^2\boldsymbol{x}_t), \frac{g_{s0}^2 g_{ts}^2}{g_{t0}^2}\boldsymbol{I}\right)\,, \tag{14}$$

We include proofs for (13) and (14) in the appendix (A.2). These equations show that we can perform inference with *any* ancestral sampling path (i.e., the timesteps can attain continuous values) by formulating the reverse process in terms of the posterior distribution as

$$p_\theta(\boldsymbol{x}_s \mid \boldsymbol{x}_t) = q\big(\boldsymbol{x}_s \mid \boldsymbol{x}_t, \hat{\boldsymbol{x}}_0 = \tfrac{1}{f_{t0}}(\boldsymbol{x}_t - g_{t0}\boldsymbol{\epsilon}_\theta(\boldsymbol{x}_t, t))\big), \tag{15}$$

justifying the compatibility of our main approach with time-continuous DDPMs. We note that this reverse process is also mathematically equivalent to a reverse process based on a time-discrete DDPM derived from a subsequence of the original timesteps as done by Song et al. (2020); Nichol & Dhariwal (2021). For the case of $s = 0$ in the reverse process, we follow the parametrization of Ho et al. (2020) to obtain discretized log likelihoods and compare our log likelihoods fairly with prior work.

Algorithm 1: Given a matrix $L \sim (T+1) \times (T+1)$ of precomputed $L(\cdot, \cdot)$ terms, find the likelihood-optimal schedules for all step budgets.

```python
def vectorized_dp_all_budgets(L):
  T = len(L) - 1
  D = np.full(L.shape, -1)
  C = np.full(L.shape, np.inf)
  C[0, 0] = 0
  for k in range(1, T + 1):
    bpds = C[k - 1, None] + L
    C[k] = np.amin(bpds, axis=-1)
    D[k] = np.argmin(bpds, axis=-1)
  return D
```

Algorithm 2: Fetch the shortest path of $K$ steps from the dynamic programming results implicitly returned by Algorithm 1.

```python
def fetch_shortest_path(D, K):
  optpath = []
  t = K
  for k in reversed(range(K)):
    optpath.append(t)
    t = D[k, t]
  return optpath
```

## 4 LEARNING TO EFFICIENTLY SAMPLE FROM DDPMS

We now introduce our dynamic programming (DP) approach. In general, after training a DDPM, one can use a different inference path than the one used during training. Additionally, one can optimize a loss or reward function with respect to the timesteps themselves *after* the DDPM is trained. In this paper, we use the ELBO as our loss function, however we note that it is possible to directly optimize the timesteps with other objectives.

### 4.1 OPTIMIZING THE ELBO

In our work, we choose to optimize ELBO as our objective. We rely on one key property of ELBOs, their *decomposability*. Before defining decomposability, we formally define a $K$-step inference path as a finite, monotonically increasing sequence of timesteps $0 = t'_0 < t'_1 < ... < t'_{K-1} < t'_K = 1$. Now, given a set $S \subseteq [0, 1]$, we define a family of lower bounds $\mathcal{L}$ of an "ideal" objective $L_{\text{ideal}}$ to be *S-decomposable* if:

1. There is a bijection from $\mathcal{L}$ to the set of all inference paths $\boldsymbol{t}$ with all timesteps in $S$, and
2. There exists a function $L : S \times S \to [0, \infty)$ such that, for all inference paths $\boldsymbol{t}$ with all timesteps in $S$, $L_{\text{ideal}} \geq \sum_{i=1}^{|\boldsymbol{t}|-1} L(t_i, t_{i-1}) + C$ ($C$ a constant).

We now show that DDPM ELBOs are decomposable. As shown by Song et al. (2020); Nichol & Dhariwal (2021) and the equations in Section 3, for any $K$ and any $K$-step inference path $\boldsymbol{t}$, there is a corresponding ELBO

$$- L_{\text{ELBO}} = \mathbb{E}_q D_{\text{KL}}\big(q(\boldsymbol{x}_1|\boldsymbol{x}_0)\|p_\theta(\boldsymbol{x}_1)\big) + \sum_{i=1}^{K} L(t'_i, t'_{i-1}) \tag{16}$$

where

$$L(t, s) = \begin{cases} -\mathbb{E}_q \log p_\theta(\boldsymbol{x}_t|\boldsymbol{x}_0) & s = 0 \\ \mathbb{E}_q D_{\text{KL}}\big(q(\boldsymbol{x}_s|\boldsymbol{x}_t, \boldsymbol{x}_0)\|p_\theta(\boldsymbol{x}_s|\boldsymbol{x}_t)\big) & s > 0 \end{cases} \tag{17}$$

Since all of these are lower bounds of $\mathbb{E}_q \log p(\boldsymbol{x}_0)$, we conclude the family of DDPM evidence lower bounds is decomposable. Specifically, a DDPM trained on a set of timesteps $S$ admits $S$-decomposable ELBOs. For DDPMs trained with continuous timesteps, $S = [0, 1]$. For DDPMs trained on discrete timesteps, $S$ is the set of those timesteps, as there is no guarantee that the behavior of the model won't be pathological when give timesteps it has never seen during training. Now the question remains, given a fixed budget $K$ steps, what is the optimal inference path?

First, we observe that any two paths that share a $(t, s)$ transition will share a common $L(t, s)$ term. We exploit this property in our dynamic programming algorithm. When given a grid $S$ of plausible inference paths $0 = t_0 < t_1 < ... < t_{T-1} < t_T = 1$ with $T \geq K$, it is possible to efficiently find the ELBO-optimal $K$-step inference path contained in $S$ by memoizing all the individual $L(t, s)$ ELBO terms for $s, t \in \{t_0, ..., t_T\}$ with $s < t$. We can then solve the canonical least-cost-path problem on a directed graph where $s \to t$ are nodes and the edge connecting them has cost $L(t, s)$.

## 4.2 Dynamic Programming Algorithm

We now outline our methodology to solve the least-cost-path problem. Our solution is similar to Dijkstra's algorithm, but it differs to the classical least-cost-path problem where the latter is typically used, as our problem has additional constraints: we restrict our search to paths of exactly $K + 1$ nodes, and the start and end nodes are fixed.

Let $C$ and $D$ be $(K + 1) \times (T + 1)$ matrices. $C[k, t]$ will be the total cost of the least-cost-path of length $k$ from $t$ to 0. $D$ will be filled with the timesteps corresponding to such paths; i.e., $D[k, t]$ will be the timestep $s$ immediately previous to $t$ for the optimal $k$-step path (assuming $t$ is also part of such path).

We initialize $C[0, 0] = 0$ and all the other $C[0, \cdot]$ to $\infty$ (the $D[0, \cdot]$ are irrelevant, but for ease of index notation we keep them in this section). Then, for each $k$ from 1 to $K$, we iteratively set, for each $t$,

$$C[k, t] = \min_s \left( C[k - 1, s] + L(t, s) \right)$$
$$D[k, t] = \arg\min_s \left( C[k - 1, s] + L(t, s) \right)$$

where $L(t, s)$ is the cost to transition from $t$ to $s$ (see Equation 17). For all $s \geq t$, we set $L(t, s) = \infty$ (e.g., we only move backwards in the diffusion process). This procedure captures the shortest path cost in $C$ and the shortest path itself in $D$. We further observe that running the DP algorithm for each $k$ from 1 to $T$ (instead of $K$), we can extract the optimal paths for *all* possible budgets $K$. Algorithm 1 illustrates a vectorized version of the procedure we have outlined in this section, while Algorithm 2 shows how to explicitly extract the optimal paths from $D$.

## 4.3 Efficient Memoization

A priori, our dynamic programming approach appears to be inefficient because it requires computing $\mathcal{O}(T^2)$ terms (recall, as we rely on all the $L(t, s)$ terms which depend on a neural network forward pass). We however observe that a single forward pass of the DDPM can be used to compute *all* the $L(t, \cdot)$ terms. This holds true even in the case where the pre-trained DDPM *learns* the variances. For example, in Nichol & Dhariwal (2021) instead of fixing them to $\tilde{g}_{ts} = \frac{g_{ts} g_{s0}}{g_{t0}}$ as we outlined in the previous section, the forward pass itself still only depends on $t$ and not $s$, and the variance of $p_\theta(\boldsymbol{x}_s | \boldsymbol{x}_t)$ is obtained by interpolating the forward pass's output logits $\boldsymbol{v}$ with $\exp(\boldsymbol{v} \log g_{ts}^2 + (1 - \boldsymbol{v}) \log \tilde{g}_{ts}^2)$. Thus, computing the table of all the $L(t, s)$ ELBO terms only requires $\mathcal{O}(T)$ forward passes.

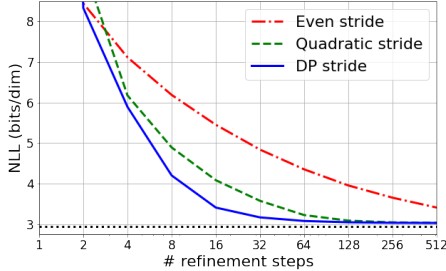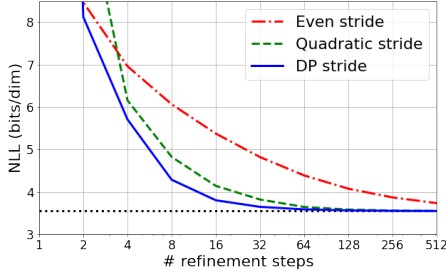

Figure 1: Negative log likelihoods (bits/dim) for $L_{\text{vlb}}$ CIFAR10 (left) and $L_{\text{hybrid}}$ ImageNet 64x64 (right) for strides discovered via dynamic programming v.s. even and quadratic strides.

## 5 Experiments

We apply our method on a wide variety of pre-trained DDPMs from prior work. This emphasizes the fact that our method is applicable to any pre-trained DDPM model. In particular, we rely the CIFAR10 model checkpoints released by Nichol & Dhariwal (2021) on both their $L_{\text{hybrid}}$ and $L_{\text{vlb}}$ objectives. We also showcase results on CIFAR10 (Krizhevsky et al., 2009) with the exact configuration used by Ho et al. (2020), which we denote as $L_{\text{simple}}$, as well as $L_{\text{hybrid}}$ on ImageNet 64x64 (Deng et al.,

Table 3: Negative log likelihoods (bits/dim) in the few-step regime across various DDPMs trained on CIFAR10, as well as state-of-the-art unconditional generative models in the same dataset. The last column corresponds to 1,000 steps for $L_{\text{simple}}$ and 4,000 steps for all other models.

| Model \ # refinement steps | 8 | 16 | 32 | 64 | 128 | 256 | All |
|---|---|---|---|---|---|---|---|
| DistAug Transformer (Jun et al., 2020) | – | – | – | – | – | – | **2.53** |
| DDPM++ (deep, sub-VP) (Song et al., 2021) | – | – | – | – | – | – | **2.99** |
| $L_{\text{simple}}$ | | | | | | | |
|    Even stride | 6.95 | 6.15 | 5.46 | 4.91 | 4.47 | 4.14 | 3.73 |
|    Quadratic stride | 5.39 | 4.86 | 4.52 | 3.84 | 3.74 | 3.73 | |
|    DP stride | 4.59 | 3.99 | 3.79 | 3.74 | 3.73 | 3.72 | |
| $L_{\text{vlb}}$ | | | | | | | |
|    Even stride | 6.20 | 5.48 | 4.89 | 4.42 | 4.03 | 3.73 | **2.94** |
|    Quadratic stride | 4.89 | 4.09 | 3.58 | 3.23 | 3.09 | 3.05 | |
|    DP stride | 4.20 | 3.41 | 3.17 | 3.08 | 3.05 | 3.04 | |
| $L_{\text{hybrid}}$ | | | | | | | |
|    Even stride | 6.14 | 5.39 | 4.77 | 4.29 | 3.92 | 3.66 | 3.17 |
|    Quadratic stride | 4.91 | 4.15 | 3.71 | 3.42 | 3.30 | 3.26 | |
|    DP stride | 4.33 | 3.62 | 3.39 | 3.30 | 3.27 | 3.26 | |

Table 4: Negative log likelihoods (bits/dim) in the few-step regime for a DDPM model trained with $L_{\text{hybrid}}$ on ImageNet 64x64 (Nichol & Dhariwal, 2021), as well as state-of-the-art unconditional generative models in the same dataset. We underline that, with just 32 steps, our DP stride achieves a score of $\leq 0.1$ bits/dim higher than the same model with the original 4,000 step budget ([*]the authors report 3.57 bits/dim, but we trained the model for 3M rather than 1.5M steps).

| Model \ # refinement steps | 8 | 16 | 32 | 64 | 128 | 256 | 4000 |
|---|---|---|---|---|---|---|---|
| Routing Transformer (Roy et al., 2021) | – | – | – | – | – | – | **3.43** |
| $L_{\text{vlb}}$ (Nichol & Dhariwal, 2021) | – | – | – | – | – | – | **3.53** |
| $L_{\text{hybrid}}$ | | | | | | | |
|    Even stride | 6.07 | 5.38 | 4.82 | 4.39 | 4.08 | 3.87 | **3.55**[*] |
|    Quadratic stride | 4.83 | 4.14 | 3.82 | 3.65 | 3.58 | 3.56 | |
|    DP stride | 4.29 | 3.80 | 3.65 | 3.59 | 3.56 | 3.56 | |

2009) following Nichol & Dhariwal (2021), training these last two models from scratch for 800K and 3M steps, respectively, but otherwise using the exact same configurations as the authors.

In our experiments, we always search over a grid that includes all the timesteps used to train the model, i.e., $\{t/T : t \in \{1, ..., T-1\}\}$. For our CIFAR10 results, we computed the memoization tables with Monte Carlo estimates over the full training dataset, while on ImageNet 64x64 we limited the number of datapoints in the Monte Carlo estimates to 16,384 images on the training dataset.

For each pre-trained model, we compare the negative log likelihoods (estimated using the full heldout dataset) of the strides discovered by our dynamic programming algorithm against even and quadratic strides, following Song et al. (2020). We find that our dynamic programming algorithm discovers strides resulting in much better log likelihoods than the hand-crafted strides used in prior work, particularly in the few-step regime. We provide a visualization of the log likelihood curves as a function of computation budget in Figure 1 for $L_{\text{simple}}$ CIFAR10 and $L_{\text{hybrid}}$ ImageNet 64x64 (Deng et al., 2009), a full list of the scores in the few-step regime in Table 1, and a visualization of the discovered steps themselves in Figure 2.

## 5.1 COMPARISON WITH FID

We further evaluate our discovered strides by reporting FID scores (Heusel et al., 2017) on 50,000 model samples against the same number of samples from the training dataset, as is standard in the literature. We find that, although our strides are yield much better log likelihoods, such optimization does not necessarily translate to also improving the FID scores. Results are included in Figure 3. This weakened correlation between log-likelihoods and FID is consistent with observations in prior work (Ho et al., 2020; Nichol & Dhariwal, 2021).

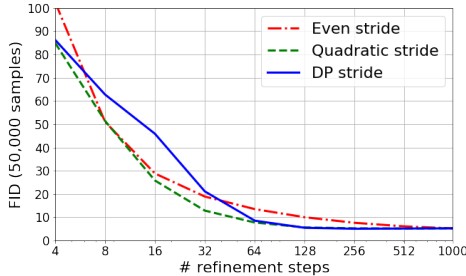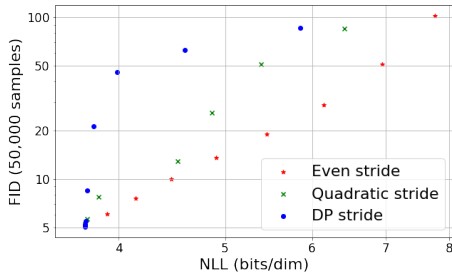

Figure 2: FID scores for $L_{\text{simple}}$ CIFAR10, as a function of computation budget (left) and negative log likelihood (right).

To remedy this issue, we show that despite our focus in this work being likelihood, we can significantly improve the FID scores discovered by our method simply by optimizing a *reweighted* ELBO. Recall that, as discussed in Section 4, our proposed method is compatible with any decomposable objective. Moreover, prior work has shown that the choice of ELBO weights has a significant effect on sample quality (Ho et al., 2020; Durkan & Song, 2021), and that choosing weights corresponds to choosing an equally valid variational lower bound of the data for a DDIM (Song et al., 2020). Similarly to prior work in the VAE literature, we thus stumble upon an open problem where different variational lower bounds compatible with the model (even with the same bits/dim) can lead to samples with different qualitative charachteristics (Alemi et al., 2018). As our focus is likelihood, we leave this research question of finding the weights / ELBO that lead to most agreement with FID for future work, but nevertheless identify one such choice that favourably biases our algorithm toward this front. Details about our construction of weights and results are included in the appendix (A.3).

## 5.2 MONTE CARLO ABLATION

To investigate the feasibility of our approach using minimal computation, we experimented with setting the number of Monte Carlo datapoints used to compute the dynamic programming table of negative log likelihood terms to 128 samples (i.e., easily fit into a single batch of GPU memory). We find that, for CIFAR10, the difference in log likelihoods is negligible, while on ImageNet 64x64 there is a visible yet slight improvement in negative log likelihood when filling the table with more samples. We hypothesize that this is due to the higher diversity of ImageNet. Nevertheless, we highlight that our procedure can be applied very quickly (i.e., with just $T$ forward passes of a neural network when using a single batch, as opposed to a running average over batches), even for large models, to significantly improve log their likelihoods in the few-step regime.

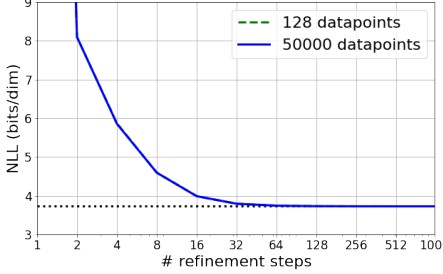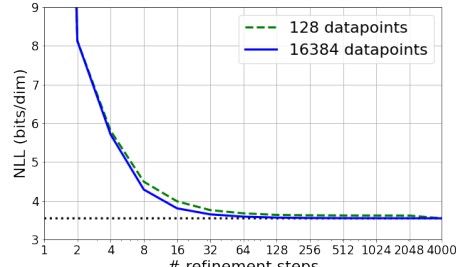

Figure 3: Negative log likelihoods (bits/dim) for $L_{\text{simple}}$ CIFAR10 and $L_{\text{hybrid}}$ ImageNet 64x64 for strides discovered via dynamic programming with log-likelihood term tables estimated with a varying number of datapoints.

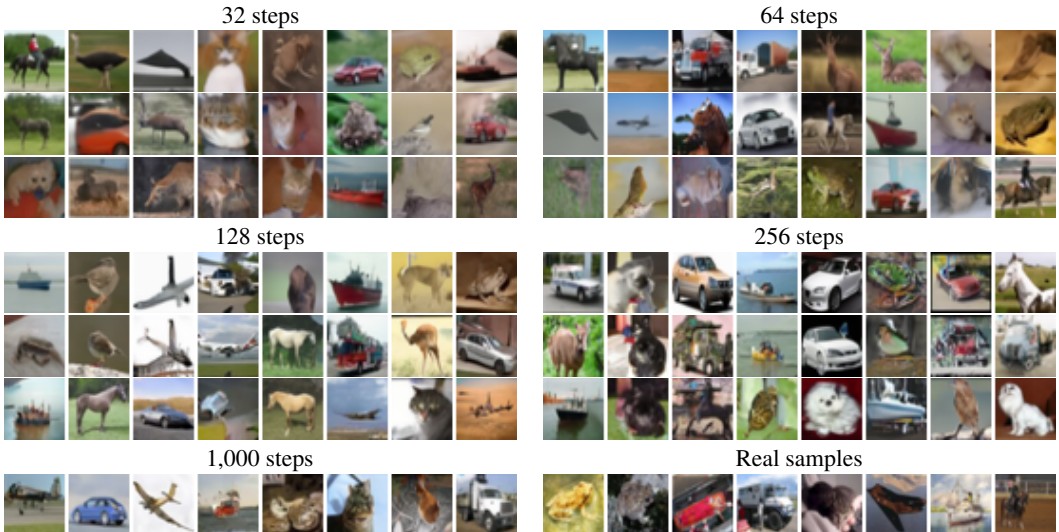

Figure 4: Non-cherrypicked $L_{\text{simple}}$ CIFAR10 samples for even (top), quadratic (middle), and DP strides (bottom), for various computation budgets. For each step budget, the samples were produced with the same random vectors.

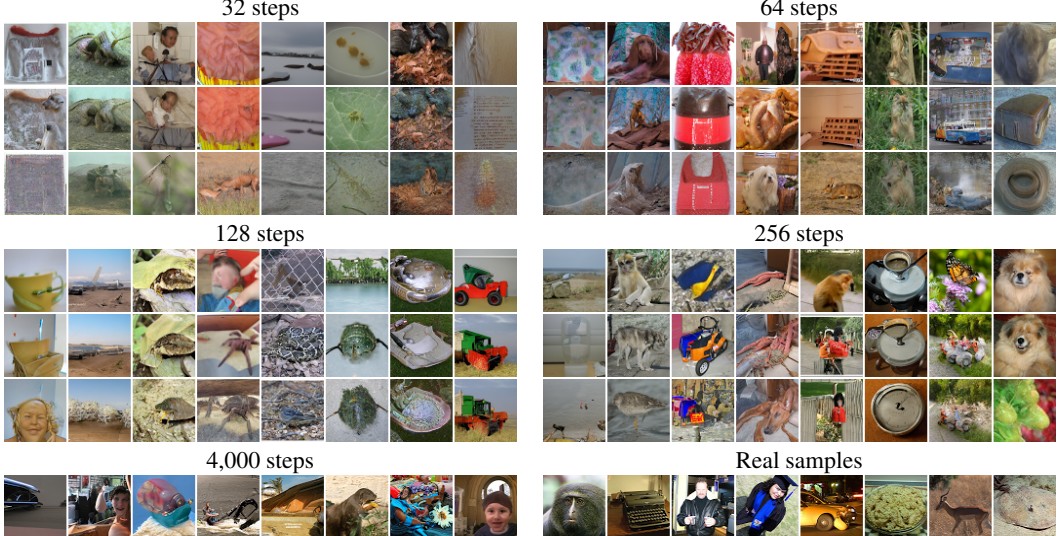

Figure 5: Non-cherrypicked $L_{\text{hybrid}}$ ImageNet 64x64 samples for even (top), quadratic (middle), and DP strides (bottom), for various computation budgets. For each step budget, the samples were produced with the same random vectors.

## 6 RELATED WORK

DDPMs (Ho et al., 2020) have recently shown results that are competitive with GANs (Goodfellow et al., 2014), and they can be traced back to the work of Sohl-Dickstein et al. (2015) as a restricted family of deep latent variable models. Dhariwal & Nichol (2021) have more recently shown that DDPMs can outperform GANs in FID scores (Heusel et al., 2017). Song & Ermon (2019) have also linked DDPMs to denoising score matching (Vincent et al., 2008; 2010), which is crucial to the continuous-time formulation (Song et al., 2021; Kingma et al., 2021). More recent work on the few-step regime of DDPMs (Song et al., 2020; Chen et al., 2021; Nichol & Dhariwal, 2021; San-Roman et al., 2021; Kong & Ping, 2021; Jolicoeur-Martineau et al., 2021) has also guided our research efforts. DDPMs are also very closely related to variational autoencoders (Kingma & Welling,

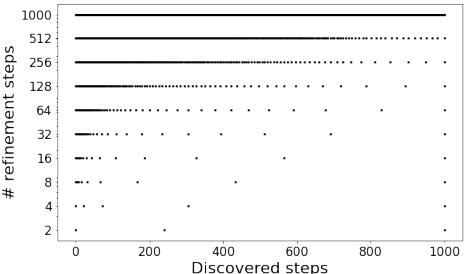 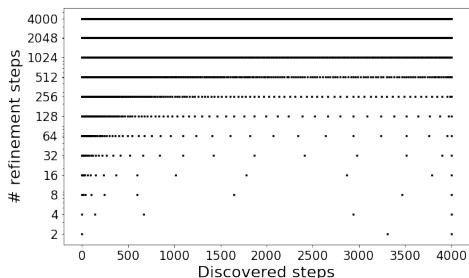

Figure 6: Timesteps discovered via dynamic programming for $L_{\text{simple}}$ CIFAR10 (left) and $L_{\text{hybrid}}$ ImageNet 64x46 (right) for various computation budgets. Each step (forward pass) is between two contiguous points. Our DP algorithm prefers allocates steps towards the end of the diffusion, agreeing with intuition from prior work where steps closer to $x_0$ are important as they capture finer image details, but curiously, it may also allocate steps closer to $x_1$, possibly to better break modes early on in the diffusion process.

2013), where more recent work has shown that, with many stochastic layers, they can also attain competitive negative log likelihoods in unconditional image generation (Child, 2020). Also very closely related to DDPMs, there has also been work on non-autoregressive modeling of text sequences that can be regarded as discrete-space DDPMs with a forward process that masks or remove tokens (Lee et al., 2018; Gu et al., 2019; Stern et al., 2019; Chan et al., 2020; Saharia et al., 2020). The UNet architecture (Ronneberger et al., 2015) has been key to the recent success of DDPMs, and as shown by Ho et al. (2020); Nichol & Dhariwal (2021), augmenting UNet with self-attention (Shaw et al., 2018) in scales where attention is computationally feasible has helped bring DDPMs closer to the current state-of-the-art autoregressive generative models (Child et al., 2019; Jun et al., 2020; Roy et al., 2021).

## 7 CONCLUSION AND DISCUSSION

By regarding the selection of the inference schedule as an optimization problem, we present a novel and efficient approach to discover a likelihood-optimal inference schedule for a pre-trained DDPM with a simple dynamic programming algorithm. Our method need only be applied once to discover the schedule, and does not require re-training the DPPM. In the few-step regime, we discover schedules on $L_{\text{simple}}$ CIFAR10 and $L_{\text{hybrid}}$ ImageNet 64x64 that require only 32 steps, yet sacrifice $\leq 0.1$ bits per dimension compared to state-of-the-art DDPMs using hundreds-to-thousands of refinement steps. Our approach only needs forward passes of the DDPM neural network to fill the dynamic programming table of $L(t, s)$ terms, and we show that we can fill the dynamic programming table with just $\mathcal{O}(T)$ forward passes. Moreover, we can estimate the table using only 128 Monte Carlo samples, finding this to be sufficient even for datasets such as ImageNet with high diversity. Our method achieves strong likelihoods with very few refinement steps, outperforming prior work utilizing hand-crafted strides (Ho et al., 2020; Nichol & Dhariwal, 2021).

Despite very strong log-likelihood results, we observe that maximizing the *unweighted* ELBO can actually lead to higher (worse) FID scores, and on ImageNet 64x64, a decrease in sample quality for the smallest budgets $K \in \{32, 64\}$; this is consistent with findings in prior work (Ho et al., 2020; Nichol & Dhariwal, 2021). Nevertheless, our method is compatible with any decomposable objective such as reweighted variational lower bounds, and we show that a simple choice of reweighted ELBO (or equivalently a choice of DDIM) can remedy this issue. Developing principled methods to choose variational lower bounds or other decomposable metrics that correlate best with image quality is thus an important direction for future research. Finally, we remark that likelihood optimization itself is useful for specific applications: as well as better compression, in domains such as non-autoregressive text generation where likelihood correlates much better with sample quality and where diffusion models are starting to make progress (Austin et al., 2021), our method has the potential to improve sampling speed with far less cost in generation fidelity and without such adaptations.

## REPRODUCIBILITY STATEMENT

We will fully open source our work and provide code pointers in the paper in the camera-ready version. Nevertheless, we provide pseudocode with a complete implementation of our proposed algorithm to maximize ease of reproducibility while we work on open-sourcing our work (see Algorithms **??** and **??**). Since we experiment with open-sourced datasets and pre-trained models that already have publicly available checkpoints, our work is fully reproducible. We additionally emphasize that our method has no hyperparameters of its own.

## ETHICS STATEMENT

Innovations in generative models have the potential to enable harmful and unethical applications. In applications where no harm is intended, bias and other failure modes of generative models and datasets used to train them can also lead to issues with fairness, discrimination, and other forms of harm. While our work is focused on making diffusion models more efficient, we believe its public release will not cause any form of immediate harm, as much more efficient generative models for images like GANs can still achieve high sample quality at a fraction of the speed of diffusion models.

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

## A APPENDIX

### A.1 PROOF FOR EQUATION 12

From Equation 10, we get by implicit differentiation that

$$f(t) = \psi(t,0) = \exp\left(\int_0^t f_{\text{sde}}(u)du\right)$$

$$\Rightarrow f'(t) = \exp\left(\int_0^t f_{\text{sde}}(u)du\right)\frac{d}{dt}\int_0^t f_{\text{sde}}(u)du = f(t)f_{\text{sde}}(t)$$

$$\Rightarrow f_{\text{sde}}(t) = \frac{f'(t)}{f(t)}$$

Similarly as above and also using the fact that $\psi(t,s) = \frac{\psi(t,0)}{\psi(s,0)}$,

$$g(t)^2 = \int_0^t \psi(t,u)^2 g_{\text{sde}}(u)^2 du = \int_0^t \frac{f(t)^2}{f(u)^2} g_{\text{sde}}(u)^2 du = f(t)^2 \int_0^t \frac{g_{\text{sde}}(u)^2}{f(u)^2} du$$

$$\Rightarrow 2g(t)g'(t) = 2f(t)f'(t)\frac{g(t)^2}{f(t)^2} + f(t)^2\frac{d}{dt}\int_0^t \frac{g_{\text{sde}}(u)^2}{f(u)^2} du = 2f_{\text{sde}}(t)g(t)^2 + g_{\text{sde}}(t)^2$$

$$\Rightarrow g_{\text{sde}}(t) = \sqrt{2(g(t)g'(t) - f_{\text{sde}}(t)g(t)^2)}. \quad \square$$

### A.2 PROOF FOR EQUATIONS 13 AND 14

From Equation 10 and $\psi(t,s) = \frac{\psi(t,0)}{\psi(s,0)}$ it is immediate that $f_{ts}$ is the mean of $q(x_t|x_s)$. To show that $g_{ts}^2$ is the variance of $q(x_t|x_s)$, Equation 10 implies that

$$\begin{aligned}
\text{Var}[x_t|x_s] &= \int_s^t \psi(t,u)^2 g_{\text{sde}}(u)^2 du \\
&= \int_0^t \psi(t,u)^2 g_{\text{sde}}(u)^2 du - \int_0^s \psi(t,u)^2 g_{\text{sde}}(u)^2 du \\
&= g(t)^2 - \psi(t,0)^2 \int_0^s \frac{\psi(s,u)^2}{\psi(s,u)^2\psi(u,0)^2} g_{\text{sde}}(u)^2 du \\
&= g(t)^2 - \psi(t,0)^2 \int_0^s \frac{\psi(s,u)^2}{\psi(s,0)^2} g_{\text{sde}}(u)^2 du \\
&= g(t)^2 - \psi(t,s)^2 g(s)^2 \\
&= g(t)^2 - f_{ts}g(s)^2.
\end{aligned}$$

The mean of $q(x_s|x_t, x_0)$ is given by the Gaussian conjugate prior formula (where all the distributions are conditioned on $x_0$). Let $\mu = f_{ts}x_s$, so we have a prior over $\mu$ given by

$$x_s|x_0 \sim \mathcal{N}(f_{s0}x_0, g_{s0}^2 I_d) \Rightarrow \mu|x_0 \sim \mathcal{N}(f_{s0}f_{ts}x_0, f_{ts}^2 g_{s0}^2 I_d) \sim \mathcal{N}(f_{t0}x_0, f_{ts}^2 g_{s0}^2 I_d),$$

and a likelihood with mean $\mu$

$$x_t|x_s, x_0 \sim x_t|x_s \sim \mathcal{N}(f_{ts}x_s, g_{ts}^2 I_d) \Rightarrow x_t|\mu, x_0 \sim x_t|\mu \sim \mathcal{N}(\mu, g_{ts}^2 I_d).$$

Then it follows by the formula that $\mu|x_t, x_0$ has variance

$$\text{Var}[\mu|x_t, x_0] = \left(\frac{1}{f_{ts}^2 g_{s0}^2} + \frac{1}{g_{ts}^2}\right)^{-1} = \left(\frac{g_{ts}^2 + f_{ts}^2 g_{s0}^2}{f_{ts}^2 g_{s0}^2 g_{ts}^2}\right)^{-1} = \frac{f_{ts}^2 g_{s0}^2 g_{ts}^2}{g_{ts}^2 + f_{ts}^2 g_{s0}^2}$$

$$\Rightarrow \text{Var}[x_s|x_t, x_0] = \frac{1}{f_{ts}^2}\text{Var}[\mu|x_t, x_0] = \frac{g_{s0}^2 g_{ts}^2}{g_{ts}^2 + f_{ts}^2 g_{s0}^2} = \frac{g_{s0}^2 g_{ts}^2}{g_{t0}^2} = \tilde{g}_{ts}^2$$

Table 5: FID scores in the few-step regime across DDPMs trained on CIFAR10 and ImageNet 64x64, compared to DDPM and DDIM($\eta = 0$) with different strides. The last column corresponds to 1,000 steps for CIAFR10 $L_{\text{simple}}$ and 4,000 steps for ImageNet 64x64 $L_{\text{hybrid}}$. The best results are highlighted in bold, and the second best are underlined.

| Model \ # refinement steps | 8 | 16 | 32 | 64 | 128 | 256 | All |
|---|---|---|---|---|---|---|---|
| CIFAR10 $L_{\text{simple}}$ | | | | | | | |
|    Even stride (DDPM) | 51.04 | 28.83 | 20.85 | 13.50 | 10.01 | 7.61 | **3.17** |
|    Quadratic stride (DDPM) | 51.44 | 25.80 | 12.85 | 7.73 | 5.63 | 5.25 | |
|    Even stride (DDIM) | 26.96 | 15.88 | 11.12 | 8.40 | 6.61 | 5.50 | |
|    Quadratic stride (DDIM) | **19.24** | **9.49** | **6.16** | **5.09** | **4.70** | **4.57** | |
|    DP stride | 62.79 | 45.90 | 21.11 | 8.52 | 5.47 | 5.05 | |
|    DP stride + MSE reweighting | 58.11 | 29.44 | 12.07 | 6.74 | 5.24 | 5.13 | |
| ImageNet 64x64 $L_{\text{hybrid}}$ | | | | | | | |
|    Even stride (DDPM) | 72.76 | 37.25 | 21.83 | **16.66** | **14.99** | **14.72** | 3.19 |
|    Quadratic stride (DDPM) | 223.6 | 73.10 | 38.69 | 22.45 | 16.99 | 15.37 | |
|    Even stride (DDIM) | **52.37** | **27.02** | **20.25** | 17.85 | 16.99 | 16.51 | |
|    Quadratic stride (DDIM) | 243.1 | 59.80 | 29.27 | 20.80 | 17.82 | 16.81 | |
|    DP stride | 184.0 | 124.0 | 71.15 | 40.45 | 23.87 | 17.62 | |
|    DP stride + MSE reweighting | 146.4 | 85.42 | 48.10 | 28.88 | 20.45 | 17.10 | |

and mean

$$\mathbb{E}[\mu|x_t, x_0] = \left(\frac{1}{f_{ts}^2 g_{s0}^2} + \frac{1}{g_{ts}^2}\right)^{-1}\left(\frac{f_{t0}x_0}{f_{ts}^2 g_{s0}^2} + \frac{x_t}{g_{ts}^2}\right) = \frac{f_{t0}g_{ts}^2 x_0 + f_{ts}^2 g_{s0}^2 x_t}{g_{ts}^2 + f_{ts}^2 g_{s0}^2} = \frac{f_{t0}g_{ts}^2 x_0 + f_{ts}^2 g_{s0}^2 x_t}{g_{t0}^2}$$

$$\Rightarrow \mathbb{E}[x_s|x_t, x_0] = \frac{1}{f_{ts}}\mathbb{E}[\mu|x_t, x_0] = \frac{\frac{f_{t0}}{f_{ts}}g_{ts}^2 x_0 + f_{ts}g_{s0}^2 x_t}{g_{t0}^2} = \frac{f_{s0}g_{ts}^2 x_0 + f_{ts}g_{s0}^2 x_t}{g_{t0}^2} = \tilde{f}_{ts}(x_t, x_0). \quad \square$$

### A.3 Reweighted ELBO results

While the focus of our work is likelihood, we report FID scores for the sake of completeness, as well as to show the adaptability of our method via reweighting to focus on sample quality, as mentioned in Section 5.1.

To choose a reweighting scheme for each $L(t, s)$ term that takes into account both $t$ and $s$, Kingma et al. (2021) show that choosing discretized terms

$$L_w(t, s) = \left[-\int_s^t w(u)\text{SNR}'(u)du\right]\|\boldsymbol{x}_0 - \hat{\boldsymbol{x}}_0(\boldsymbol{x}_t, t)\|^2 \tag{18}$$

ensures that at the limit of infinite timesteps the continuous ELBO becomes $-\frac{1}{2}\mathbb{E}_q\int_0^1 w(t)\text{SNR}'(t)\|\boldsymbol{x}_0 - \hat{\boldsymbol{x}}_0(\boldsymbol{x}_t, t)\|^2 dt$ (where $\text{SNR}(t) = \frac{f_{t0}^2}{g_{t0}^2}$, and a constant $w(t) = 1$ yields the unweighted ELBO).

While choosing $w(t) = -\frac{\text{SNR}(t)}{\text{SNR}'(t)}$ (which is $L_{\text{simple}}$ at the limit) did not work, we find that choosing $w(t) = -\frac{1}{\text{SNR}'(t)}$ (which leads to a continuous objective of unweighted mean square errors and $L_w(t, s) = (t - s)\|\boldsymbol{x}_0 - \hat{\boldsymbol{x}}_0(\boldsymbol{x}_t, t)\|^2$) allows our algorithm to outperform DDPM FID scores and achieve similar scores to DDIM (Song et al., 2020). We call this "MSE reweighting" and include FID scores in Table 5, comparing to DDPM but also DDIM($\eta = 0$) which does not admit likelihood computation but has been shown to be one of the strongest FID baselines in the few-step regime. The FID scores were estimated with 50,000 model and training data samples, as is standard in the literature.

### A.4 NOTE ON SAMPLING STRATEGIES FOR FAIR COMPARISON

Finally, we discuss an alternative approach to that used in the figures of the paper for producing more qualitatively comparable samples. Given a fixed budget $K$, Figures 4 and 5 are produced to generate "comparable" samples across different strides by fixing all the standard Gaussian vectors in the sampling chain. However, another approach that allows to compare samples across *different* budgets is to fix a single Brownian motion trajectory on all $T$ steps, and using discretizations (based on any stride) of this single random trajectory to generate the samples. We empirically find, however, that the former approach tends to produce much more similar images (see Figures 7 and 8 below). We suspect that the use of different strides (and hence different random directions from the fixed random trajectory) along with the very chaotic, non-linear behavior of DDPMs is the cause of this behavior.

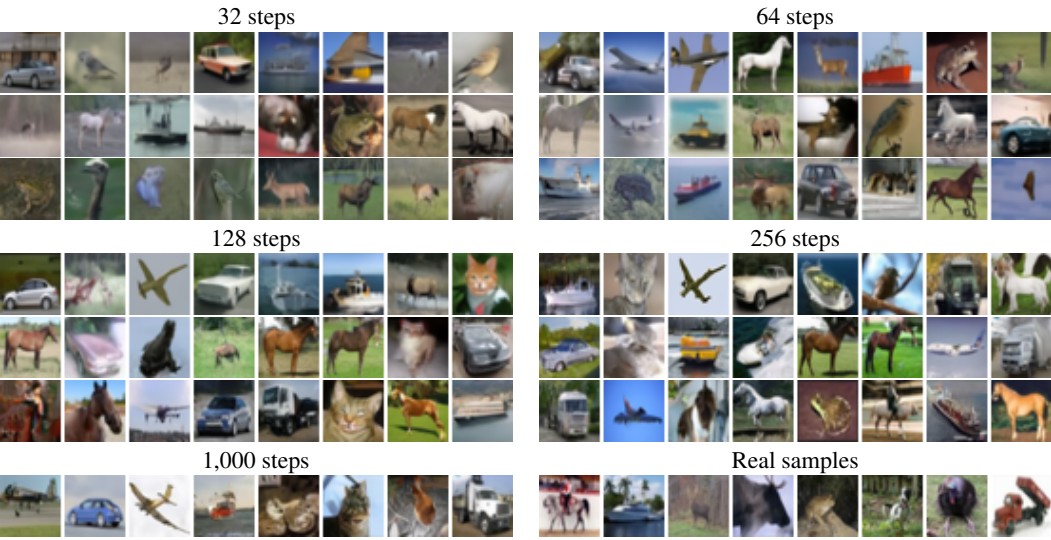

Figure 7: Non-cherrypicked $L_{\text{simple}}$ CIFAR10 samples for even (top), quadratic (middle), and DP strides (bottom), for various computation budgets. All samples use the same random trajectory.

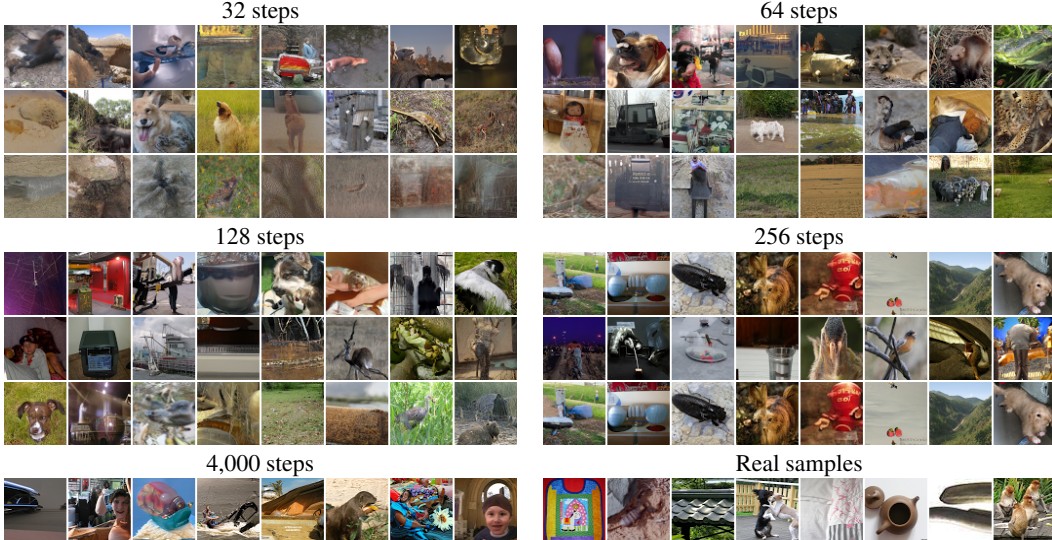

Figure 8: Non-cherrypicked $L_{\text{hybrid}}$ ImageNet 64x64 samples for even (top), quadratic (middle), and DP strides (bottom), for various computation budgets. All samples use the same random trajectory.

