# OpenReview forum: "Learning to Efficiently Sample from Diffusion Probabilistic Models"
_ICLR.cc/2022/Conference — ICLR 2022 Submitted_

### Official Review · Reviewer_2CEn · 2021-10-17

**Correctness:** 4
**Technical Novelty And Significance:** 2
**Empirical Novelty And Significance:** 4
**Recommendation:** 6
**Confidence:** 4

**Main Review:**

The paper is very clearly written, and I enjoyed reading this paper.

The main attraction of this paper is the dramatic increase in computational efficiency -- the authors discuss one example in which 4000 steps in the SDE solver are replaced with merely 32 steps. [I will tend to refer to things as being steps of an SDE solver, even in the discrete case, since that's basically what's going on.] This is certainly a dramatic claim, as the high computational cost of DDPMs has so far been one of their major limiting factors.

The suggested technique seems to be mostly reasonable. Overall the dramatic reduction in steps feels "too good to be true" -- a sentiment that is largely borne out by Section 5.1, in which it is demonstrated that improving the ELBO does not necessarily imply improving the FID. As the authors note, it is multiple to derive multiple valid ELBOs, so this is a case in which optimising the ELBO need not imply actually improving the model.

Overall, my take on this paper is that speed is improved, but it is hit-and-miss whether model performance is compromised whilst doing so. This is reflected in my middling-acceptance score. With some refinement I could see the techniques this paper proposes being of great utility.

## Figure 5

One meaningful weakness in the presentation is Figure 5, in which I think different Brownian sample paths were used to generate each image. I do note that the text claims that the same random seed was used, but the variety -- both within each group-of-steps, and between each group-of-steps, means I am skeptical. My guess is that (a) different Brownian sample paths were used for each group of steps, and (b) within each group of steps, "using the same random seed" does not actually refer to using the same Brownian motion; rather it refers to using the same increments (each of which are rescaled by $\alpha$, $\sigma$ or $g$, depending on your notation). This is not at all the same thing as using the same Brownian motion.

The appropriate thing to do would be to use the same continuous-time Brownian motion sample for every single picture shown in Figure 5. Every time a point is queried (presumably nearly always at a point that it has not been queried at before, as different step schemes may place steps is very different places), then a Brownian bridge should be constructed between the two samples already observed either side of it.

The authors have not released code so I cannot see what library they are using themselves, but the above procedure may easily be done using the `BrownianInterval` of the `torchsde` library [1]. Make sure to use a single `BrownianInterval` object for the entirety of generating a figure (recreating a new one at any point would be a mistake, as it is deterministic only up to both its seed *and* the points it has already been queried at). (To give the appropriate references: the "Brownian Interval" was introduced in [2], as an improvement of the "Virtual Brownian Tree" of [3].)

If the above procedure is followed then I would expect the generated samples to much more closely resemble each in other, and in doing so be able to better understand the effect of increasing the number of steps. (Which is, after all, central to this paper.)

## Other remarks

Equation (16) is clearly central to the paper. However, it pretty much comes out of nowhere. (At least for the reader who doesn't hold all the mathematics of DDPMs in their head.) I think that a derivation would be a meaningful improvement to the paper.

The dynamic programming algorithm outlined in Section 4.2 feels essentially standard -- besides Dijkstra's algorithm, it also seems very reminiscent of dynamic time warping. I regard the main contribution of this paper as the identification that step locations can be chosen via DP; not the algorithm itself.

The entire paper is framed only in the context of inference. I speculate that it might also be useful in the context of training: minimising training costs, especially for expensive models such as these, is a topic of great importance. Perhaps the procedure suggested in this paper could be re-run every N training steps, for some N?

## Ethics statement

I would have thought that improving the computational efficiency of costly models would have some (perhaps small) positive impact on the pressing issue of climate change. It seems a bit perverse that this *positive* ethical impact is not discussed in the ethics statement.

## Minor points

- $D_{KL}$ never has brackets around its arguments -- e.g. it's just $D_{KL} p(x)q(x)$ rather than $D_{KL}(p(x), q(x))$ or $D_{KL}(p(x)||q(x))$.
- Page 4: The abbreviation "i.e." is usually discouraged in academic writing.
- Algorithms 1 and 2: These are a weird a mix of pseudocode and Python. I think it would be preferred to pick just one. (Especially as they rely on behaviour specific to NumPy, such as indexing by `None`.)
- I am not convinced how meaningful the discussion in Section 4.3 really is. It points out that $O(T)$ forward passes are required. As each forward pass takes $O(T)$ work then overall $O(T^2)$ work is required -- exactly as expected. What is new here?
- I don't think "BPD" (page 7) is defined.

## References

[1] Li. "torchsde" https://github.com/google-research/torchsde

[2] Kidger et al. "Efficient and Accurate Gradients for Neural SDEs" NeurIPS 2021 https://arxiv.org/abs/2105.13493

[3] Li et al. "Scalable Gradients for Stochastic Differential Equations" AISTATS 2020 https://arxiv.org/abs/2001.01328

**Summary Of The Paper:**

Samples are generated from DDPMs by solving an SDE (often in "discrete time", which is used to refer to specifically the Euler--Maruyama discretisation). This necessitates a choice for where to make numerical steps. Each choice of step locations has a corresponding ELBO. This paper demonstrates that (on a pretrained model) the optimal ELBO may be obtained via a dynamic programming algorithm for the location of the steps.

**Summary Of The Review:**

Possibly with some refinement, the paper has the potential to be very good. As it stands it presents a dramatic speed improvement that may-or-may-not produce compromise the final model. Overall I recommend acceptance.

---

> ### Author Response · Authors · 2021-11-13
> **Reply to reviewer 2CEn**
>
> Thank you for your valuable review and feedback. We address your specific concerns below.
>
> > Overall the dramatic reduction in steps feels "too good to be true" -- a sentiment that is largely borne out by Section 5.1, in which it is demonstrated that improving the ELBO does not necessarily imply improving the FID. As the authors note, it is multiple to derive multiple valid ELBOs, so this is a case in which optimizing the ELBO need not imply actually improving the model.
>
> As we discuss in the paper and provide results in the appendix, this rate-distortion issue can be remedied by choosing a different ELBO, and doing so in a principled manner is important future work. As this concern is also shared by reviewer xxjp, we encourage the reviewer to also read that reply, as well as our general comment where we address this specific issue in more depth.
>
> > One meaningful weakness in the presentation is Figure 5, in which I think different Brownian sample paths were used to generate each image. I do note that the text claims that the same random seed was used, but the variety -- both within each group-of-steps, and between each group-of-steps, means I am skeptical.
>
> We use JAX, which uses a deterministic procedure to split PRNG keys. This means all images (for a fixed step budget) use the same random Gaussians vectors at each step of the sampling chain. It is still possible for different schedules to end up in different modes if a particular schedule leads an early step to a slightly different point, as all the modes are brought together the higher the noise level.
>
> > The entire paper is framed only in the context of inference. I speculate that it might also be useful in the context of training: minimising training costs, especially for expensive models such as these, is a topic of great importance. Perhaps the procedure suggested in this paper could be re-run every N training steps, for some N?
>
> Concurrent work to ours, such as Variational Diffusion Models ([Kingma et al., 2021](https://arxiv.org/abs/2107.00630)), might be a more natural way to find schedules with better log-likelihoods during training. Our procedure however can efficiently get the optimal schedules for \textit{all} budgets, while gradient-based optimization requires fixing a single step budget.
>
> > I am not convinced how meaningful the discussion in Section 4.3 really is. It points out that $\mathcal{O}(T)$  forward passes are required. As each forward pass takes $\mathcal{O}(T)$ work then overall $\mathcal{O}(T^2)$ work is required -- exactly as expected. What is new here?
>
> We do need to pre-compute $\mathcal{O}(T^2)$ ELBO terms, but to do this, we only need each $\mathcal{O}(T)$ forward passes as mentioned in the paper (remember, the forward pass required to estimate $L(t,s)$ only depends on $t$, so it can be reused to estimate multiple terms, see eqn 15). Since the forward pass is the computational bottleneck, this note is important for practitioners to efficiently implement our procedure.
>
> We also agree with the rest of the feedback– especially the points relating to clarity and writing– which we have omitted for brevity but factored it all into the paper. Thank you!

---

> > ### Comment · Reviewer_2CEn · 2021-11-13
> > **Response**
> >
> > > We use JAX, which uses a deterministic procedure to split PRNG keys. This means all images (for a fixed step budget) use the same random Gaussians vectors at each step of the sampling chain. It is still possible for different schedules to end up in different modes if a particular schedule leads an early step to a slightly different point, as all the modes are brought together the higher the noise level.
> >
> > Right -- *for a fixed step budget*. My point is that the same single continuous time Brownian sample path should be used for every image in Figure 4. (And another single fixed Brownian sample path for the entirety of Figure 5.) This is independent of the number of steps or their locations.
> >
> > Doing so would make it possible to visually compare the difference in quality between steps.
> >
> > > We do need to pre-compute $\mathcal{O}(T^2)$ ELBO terms, but to do this, we only need each $\mathcal{O}(T)$ forward passes as mentioned in the paper (remember, the forward pass required to estimate $L(t, s)$ only depends on $t$, so it can be reused to estimate multiple terms, see eqn 15). Since the forward pass is the computational bottleneck, this note is important for practitioners to efficiently implement our procedure.
> >
> > My point is that I think this is obvious, and that this paragraph could be cut.

---

> > > ### Author Response · Authors · 2021-11-19
> > > **On comparable samples across different budgets**
> > >
> > > Thank you for clarifying this. We sampled our models with this strategy, fixing the entire random trajectory for all step budgets, and observed that the images actually do not look as similar as they do with our previous approach (though some backgrounds / object classes / etc. do seem to align sometimes). We suspect this is because different strides discretize the fixed trajectory in different ways, while with our previous approach, at a fixed budget the different strides used the same noise vectors. E.g., with this new approach, picking all even steps v.s. all odd steps would result in completely different samples, despite the trajectory being fixed and the strides being *almost* identical for practical purposes. We have nevertheless added a brief discussion about these alternative “fair comparison” sampling strategies in the appendix, including the new samples with the single, fixed random trajectory that you have suggested.

---

### Official Review · Reviewer_EgPP · 2021-10-28

**Correctness:** 4
**Technical Novelty And Significance:** 2
**Empirical Novelty And Significance:** 2
**Recommendation:** 5
**Confidence:** 3

**Main Review:**

Some things are introduced but never clearly explained. The most important one is you never explicitly say what *decomposable* means. I think most readers can figure it out, but only *after* reading the paper. You write a bit on page 2, but I would make it even more clear, as it is important for reading the rest.
Also: what is an ELBO path?

On page 4, you write: "we can optimize a loss or reward function with respect to the timesteps themselves (after the DDPM is trained)."
Can you explain again what this means?

Condition 1 on page 4: "The path starts at t = 0 and ends at t = 1."
Is it not possible to both scale and translate the timescale? How restrictive is this really?

What is it about some regularization methods that makes your approach not work? Breaking the decomposability?
Can the authors think of other regularization methods that break the approach?

My key concern: In *actual* compute time (say, seconds) how long does it take DP stride with 128 steps take compared to 128 Quadratic stride? To me it looks like 128 is enough for quadratic stride to catch up to your method, so how much is there to win by choosing your algorithm?

**Summary Of The Paper:**

This paper presents a dynamic programming algorithm to sample from diffusion models. In short, they solve a Dijkstra-type problem on pretrained diffusions, and show good results even with coarse discretizations.

**Summary Of The Review:**

Overall, the paper gives a nice overview of the literature and present a new inference scheme in post-training scenarios. I have some remarks above that can make me reconsider my evaluation, and I hope for a nice discussion with the authors.

---

> ### Author Response · Authors · 2021-11-13
> **Reply to reviewer EgPP**
>
> Thank you for your valuable review and feedback. We first address your key concern:
>
> > In actual compute time (say, seconds) how long does it take DP stride with 128 steps take compared to 128 Quadratic stride? To me it looks like 128 is enough for quadratic stride to catch up to your method, so how much is there to win by choosing your algorithm?
>
> Any sampling chain with the same number of steps (e.g., 128) incurs the same running time. The difference is that we follow a different probabilistic model discovered by our approach, which results in better log-likelihood than any hand-crafted stride. If we understand the concern correctly, it is that at K≥128 the difference in BPD can become almost negligible. For many real-world applications, this budget is still too costly, so the most relevant results are for the smallest budgets, where the wins in BPD with our strides are very clear. Please let us know if we have interpreted your concern correctly! We are happy to discuss further in any case.
>
> We also agree with the clarity concerns and have incorporated your feedback into the paper. Please see our responses below.
>
> >...you never explicitly say what decomposable means.
>
> Please see our edits to subsection 4.1, where we have now included a much more precise definition for decomposability.
>
> > Also: what is an ELBO path?
>
> We agree that “ELBO path” and “inference path” were used inconsistently. We have now more clearly defined “inference paths” as ELBOs with particular choices of timesteps and made its usage more consistent throughout the paper.
>
> > Condition 1 on page 4: "The path starts at t = 0 and ends at t = 1." Is it not possible to both scale and translate the timescale? How restrictive is this really?
>
> We are using a normalized scale for notation purposes, and added this to the respective bullet point for clarification. The DDPM should be fed unnormalized timesteps, i.e., at the scale used during training.
>
> > What is it about some regularization methods that makes your approach not work? Breaking the decomposability? Can the authors think of other regularization methods that break the approach?
>
> What does the reviewer mean by “regularization approaches”? If it’s the use of ELBO reweighting, it completely depends on the choice of weights. For example, some raise the cost too much of jumps $t \to s$ near $t=0$, causing the dynamic programming solver to allocate most or all timesteps near $t=1$. A reweighted ELBO however does not break decomposability: it’s just changing $L(t,s)$ so the total cost is still a sum of individual cost terms that form a valid path (as characterized in section 4.1).

---

> > ### Author Response · Authors · 2021-11-26
> > **Re. Reply to reviewer EgPP**
> >
> > We believe we have addressed all of your comments in our response. We would appreciate it if you could acknowledge that you have read the author response. Please let us know if you have any other questions or comments!

---

> > > ### Comment · Reviewer_EgPP · 2021-11-26
> > > **Thank you**
> > >
> > > I thank the reviewers for their response and apologise my late response. I have read the response and the discussion with other reviewers as well. There were clarifications made in this process.
> > >
> > > I am maintaining my score, but I think the work has merit and I will not stand in the way of acceptance if the AC believes it should be.

---

### Official Review · Reviewer_xjpp · 2021-11-02

**Correctness:** 3
**Technical Novelty And Significance:** 3
**Empirical Novelty And Significance:** 3
**Recommendation:** 5
**Confidence:** 4

**Main Review:**

Strengths:
* The dynamic programming problem identified by the authors is an elegant and efficient approach to address the sampling limitations of DDPMs. It is natural to frame the search for an optimal schedule as a Dynamic Programming problem, and the authors show this problem can be efficiently solved in linear rather than quadratic time.
* The proposed method shows a significant improvement in model performance as measured by log likelihood compared to prior methods when applying a pre-trained DDPM over a greatly reduced set of time steps.

Weaknesses:
* The main weakness of this work is that the method appears to overfit the ELBO objective without improving (and potentially reducing) the visual quality of generated samples. In particular, the proposed method can significantly improve the log likelihood over few-step diffusion paths compared to prior techniques. However, the Dynamic Programming step schedules can actually decrease the quality of visual appearance, as measured by FID, compared to previous methods. Personally, I consider FID to be a much more reliable indicator of model quality than the log likelihood, due to its sensitivity to small changes, ability to detect mode coverage, and the fact that FID is model-agnostic, while log likelihood can only be applied to models with a tractable density or ELBO. The authors acknowledge this limitation and explore efficient schedules for maintaining low FID/high visual quality, but these results do not improve upon prior methods. Thus, while the authors achieve their intended goal of efficient and high log likelihoods via their new method, the outcome might not be particularly meaningful since it doesn't really improve model/sample quality.
* I am unsure of the relevance of Section 3. How does this fit into the presentation in Section 4? See "Other Comments" below.

Other Comments:
* In Section 3, there is a claim that "These equations show that we can perform inference with any ancestral sampling path (i.e., the timesteps can attain continuous values)" but in Section 4, there is a claim that "For time-continuous DDPMs, the choice of grid (i.e., the $t_1, \dots,  t_{T −1})$ can be arbitrary. For models trained with discrete timesteps, the grid must be a subset of (or the full) original steps used during training." Why does the method not work for arbitrary continuous time steps if the model is trained with discrete time steps? The first claim makes it seems like that would be possible.
* Why were some of the models used retrained, instead doing testing using only fixed pretrained models?

**Summary Of The Paper:**

This work presents a method to efficiently sample from a pre-trained DDPM by solving a dynamic programming problem that can maximize the log likelihood of the data samples given a fixed computational budget. This is done by defining a least-cost path problem to select a reduced set of time steps among a full grid of potential time steps across different possible step budget sizes, where the ELBO is used as the cost function. The authors show that their method can identify DDPM schedules that can achieve significantly higher log likelihood (i.e. lower bits/dim) than prior DDPM schedules in the regime where about a hundred steps or fewer are used.

**Summary Of The Review:**

Overall, I found the approach to efficient DDPM sampling employed by the authors to be sensible and reasonably novel. While their method can indeed effectively increase log likelihood for DDPM with a greatly reduced grid of time steps, this did not appear to translate to improved model quality in terms of actual generated samples. The final conclusion is therefore somewhat unsatisfying because an ideal DDPM schedule would be short and efficient, able to produce high log likelihoods, and able to produce low FID scores compared to other methods. Since this goal is not achieved, I recommend that the authors revisit their approach to identify if there is a way to more effectively incorporate sample quality (rather than log likelihood) in their DP algorithm.

---

> ### Author Response · Authors · 2021-11-13
> **Reply to reviewer xjpp**
>
> Thank you for your valuable review and feedback. Please find our responses below.
>
> >The main weakness of this work is that the method appears to overfit the ELBO objective without improving (and potentially reducing) the visual quality of generated samples.
>
> As we discuss in the paper, the disconnect between log-likelihood and sample quality can be interpreted as a rate-distortion issue, rather than an issue of overfitting. I.e., different ELBOs admit distinct sample quality, and there is a separate problem of choosing an ELBO that correlates best with human perception. Also note this problem is much less present in other domains like NLP, where likelihood correlates much better with apparent quality. Please see our general reply where we address this issue more in depth!
>
> > Why does the method not work for arbitrary continuous time steps if the model is trained with discrete time steps? The first claim makes it seems like that would be possible.
>
> A DDPM trained with fixed, discrete timesteps might learn to model the timestep information as isolated points. There is no guarantee that its behavior won’t be pathological for timesteps the model has never seen. We have added this remark to the paper for clarity.
>
> > Why were some of the models used retrained, instead doing testing using only fixed pretrained models?
>
> In our experiments, only pretrained diffusion models are used, and none of the models is re-trained. We have updated the paper to remove any source of confusion.

---

> > ### Comment · Reviewer_xjpp · 2021-11-21
> > **Response**
> >
> > I appreciate the author responses. We are mostly on in agreement except for the points regarding log likelihood, FID, and sample quality. It is clear that the DP stride achieves significantly higher log likelihood than that Quadratic stride using a budget of 32 steps. Putting aside FID for the moment, based on the results that the authors have presented, it is my subjective evaluation that the samples from the DP stride have significantly less realistic appearance than the Quadratic stride samples using a budget of 32 steps. While this is a subjective evaluation, in this case it seems clear to me. FID score backs up the visual evaluation. By the time 64 steps are used, the samples presented by the authors look about the same to me for DP and Quadratic stride, which is again backed up by FID. The fundamental issue is that sample quality is not improved by the DP stride, which both visual inspection and FID support. This happens even though log likelihood is higher for the DP method. This is what makes me claim (perhaps in a non-rigorous sense) that the proposed method is "overfitting" to one metric of generative modeling (log likelihood on the test set) while missing broader generative modeling goals (as measured by human assessment of sample quality, FID, etc.).
> >
> > To elaborate further, I do not believe that log likelihood is nearly is rigorous as the authors suggest, and I also believe that FID is much less susceptible to the kind of "overfitting" that is occurring in this situation. It is known, counter-intuitively, that achieving a high log likelihood on a test set indicates neither correct density estimation nor high visual quality of samples [a]. This is because the incorrect mixture model $p_\theta (x) \approx \alpha q(x) + (1 - \alpha) u (x)$, where $q(x)$ is the data distribution and $u(x)$ is a noise distribution, has nearly the same log likelihood as a correct model $p_\theta (x) \approx q(x)$, even when $\alpha$ is extremely small (for example, $\alpha = 10^{-10}$). This is discussed in [a], "Great Log Likelihood and Poor Samples". Although this is speculative, I suggest that the DP method might be exploiting such a shortcut to aggressively maximize log likelihood at the expense of image quality.
> >
> > If FID were used as a differentiable objective function instead, it might be the case that the samples achieve very low FID but still have poor visual quality and low log likelihood, since Inception Score and FID can still be attacked using adversarial methods [b]. However, FID is not used directly as a learning metric, while log likelihood is used directly, which would seem to make "overfitting" to log likelihood more of a practical concern. Early autoregressive image models were the most effective at the time for maximizing log likelihood but the synthesis results tended to lag behind other model families, perhaps due to a similar "overfitting" of the log likelihood objective. Although log likelihood is more calibrated with sample quality in the NLP domain, most models in the NLP domain are autoregressive and trained to optimize the log likelihood directly. It is perhaps unsurprising that the relative performance of models trained with the same metric would translate to similar relative performance across other metrics. Image modeling has a broader array of methods and I would argue FID is a more reliable measure of performance. The objections to FID raised in [c] do not really apply here because [c] discusses comparison of FID scores across different frameworks. Within a given framework, it is my experience that the relative values of FID scores are indicative of relative performance. Since the authors calculated all scores in the same framework, the issue in [c] does not apply.
> >
> > Given the overly-aggressive optimization at the expense of visual quality, the paper seems to be missing a satisfying conclusion. I further do not feel that log likelihood should be presented as better simply because it is statistically grounded, since it has some very counter-intuitive properties. The authors themselves suggest two directions for further investigations that I feel would make their work a more complete product. These are: 1) tune DP strides to give both good log likelihood and sample quality or 2) demonstrate a situation where aggressive log likelihood optimization alone is enough to achieve the desired goals. Two ideas for 2) raised by the authors are a compression task and a language modeling task. While I see the value in this work, I would like to be convinced that the DP stride method can either achieve a more well-rounded results across different metrics or that aggressively optimizing log likelihood alone is useful for the model.
> >
> > [a] A note on the evaluation of generative models. https://arxiv.org/pdf/1511.01844.pdf
> > [b] A note on inception score. https://arxiv.org/pdf/1801.01973.pdf
> > [c] Surprising Subtleties in FID Calculation. https://arxiv.org/pdf/2104.11222.pdf

---

> > > ### Author Response · Authors · 2021-11-26
> > > **Re. Response**
> > >
> > > Thanks for taking the time to draft this thorough response.
> > >
> > > We agree that FID and subjective evaluation of samples are important metrics for evaluating image synthesis. That said, memorization of the training examples can yield near-optimal FID and subjective evaluation scores. A comprehensive discussion of the best way to evaluate generative models (FID vs log-likelihood vs some other metrics) is beyond the scope of this paper. Our main argument is that one should pay attention to all of these metrics, as each one has its own pros and cons.
> > >
> > > The key contribution of this paper is a novel dynamic programming algorithm that finds optimal few-step diffusion samples based on log-likelihood. We believe this is a worthwhile contribution to the community, and other papers have made use of our work already. We agree that there is more research to be done in this space, especially for optimizing FID. However, we believe each individual paper should not be expected to address all the metrics for evaluating generative models.
> > >
> > > Nevertheless, we would like to point out the discussion in the appendix (A.3, Table 5), where we show that better FID than both the linear and quadratic strides can be achieved by applying the DP algorithm to a different ELBO (though we do acknowledge that these results lag behind DDIM, which is one of the strongest FID baselines for few-step DDPM sampling). Beyond the better FID scores, however, these results are a piece of evidence showing that the disconnect between log-likelihood and FID is not really an issue of overfitting (even in a non-rigorous sense), but rather, an issue of rate distortion, i.e., some ELBOs correlate better than others to sample quality. Which ELBO does this *best*, however, is an open research question that likely merits its own paper. Even DDPM training has this issue: training with $L_{\mathrm{simple}}$ leads to better quality than training with the usual ELBO, but it’s currently unknown which reweighting works best and whether there are even better ones.

---

### Author Response · Authors · 2021-11-13
**On the importance of log-likelihood**

Dear reviewers,

Thank you for your valuable feedback. Following a lot of the feedback you have all provided, we have updated the paper. Please see the revised version. In addition to this and each of the individual replies, we’d like to offer some general remarks in response to the following question:

>Why should one care about better log-likelihood if sample quality can get worse with an improvement of log-likelihood?

1. Log-likelihood is statistically grounded, whereas many recent sample quality scores are still work in progress (e.g., see [Surprising Subtleties in FID Calculation](https://arxiv.org/abs/2104.11222)). Furthermore, in domains such as NLP, log-likelihood is well-established as a reflection of human perception of quality. Concurrent work of [Austin et al. (2021)](https://arxiv.org/abs/2107.03006) has made progress in applying diffusion to text, and we suspect our work will be directly applicable here.
2. Generative models are useful beyond generating perceptually pleasing images. For an application of generative models to compression, optimizing log-likelihood is critical. Diffusion models have delivered state-of-the-art image compression results (e.g, see Variational Diffusion Models ([Kingma et al., 2021](https://arxiv.org/abs/2107.00630)). Here, log-likelihood directly captures the number of bits per dimension required for compression.
3. Subsequent work on Autoregressive Diffusion Models ([Hoogeboom et al., 2021](https://arxiv.org/abs/2110.02037)) has also shown that our approach can also be used to partially parallelize (as opposed to skip) computation by considering more weakly-conditioned ELBOs (rather than shorter ones). Given a budget, our approach yields the optimal partitioning of steps to compute in parallel.
4. The proposed dynamic programming algorithm supports any decomposable objective function, not just the DDPM ELBO.  As discussed and evaluated in the appendix of the paper, optimizing a different ELBO can lead to better FID and sample quality. We show this is true for our reweighted DDPM ELBO, which is a valid variational lower bound (see Theorem 1 in [Song et al. (2021)](https://arxiv.org/abs/2010.02502)). Finding which ELBOs correlate best with perceptual quality scores remains an important open problem.

---

> ### Author Response · Authors · 2021-11-19
> **More on the importance of log-likelihood**
>
> We agree with reviewers that most of the experiments focus on maximizing log-likelihood. But as we explained above, log-likelihood is statistically grounded and is a principled way to evaluate generative models, as it directly captures how well the generative model captures the data distribution. Sample quality scores like FID, on the other hand, rely on a classifier that has its own implicit biases. Hence, focusing too much on optimizing such scores can lead to amplifying biases present in the pre-trained classifier, whereas log-likelihood is simpler and less biased. Importantly, many existing quality scores do not reflect sample diversity very well, which is again linked to the issue of bias in generative models. Accordingly, we believe we should continue relying on log-likelihood to measure sample diversity, while paying attention to other quality scores.
>
> If any of the reviewers disagree that we should continue to track log-likelihood and try to maximize it, we are happy to engage in a constructive discussion.

---

### Decision · Program_Chairs · 2022-01-20

**Decision:**

Reject

**Comment:**

This paper proposes a dynamic programming strategy for faster approximate generation in denoising diffusion probabilistic models.

All reviewers appreciated the paper, but they are not overly excited.

Two reviewers are focused on the log likelihood not being the objective for image quality. This AC does not really buy this argument.

The method and story around are well-rounded and finished. So it is hard to think of any major modifications that will change the overall story a lot. One could therefore argue for acceptance as it stands. On the other hand this is difficult to argue for given the below acceptance level scores.

So the final recommendation is reject with a strong encouragement to submit to the next conference. Updating the paper with preemptive arguments on why the ELBO and not FID is the right thing to consider.